# Learning 3D Persistent Embodied World Models

**Siyuan Zhou**[*]   **Yilun Du**[‡]   **Yuncong Yang**[†]   **Lei Han**[§]
**Peihao Chen**[§]   **Dit-Yan Yeung**[*]   **Chuang Gan**[†]

## Abstract

The ability to simulate the effects of future actions on the world is a crucial ability of intelligent embodied agents, enabling agents to anticipate the effects of their actions and make plans accordingly. While a large body of existing work has explored how to construct such world models using video models, they are often myopic in nature, without any memory of a scene not captured by currently observed images, preventing agents from making consistent long-horizon plans in complex environments where many parts of the scene are partially observed. We introduce a new persistent embodied world model with an explicit memory of previously generated content, enabling much more consistent long-horizon simulation. During generation time, our video diffusion model predicts RGB-D video of the future observations of the agent. This generation is then aggregated into a persistent 3D map of the environment. By conditioning the video model on this 3D spatial map, we illustrate how this enables video world models to faithfully simulate both seen and unseen parts of the world. Finally, we illustrate the efficacy of such a world model in downstream embodied applications, enabling effective planning and policy learning.

## 1   Introduction

By training on vast and diverse datasets from the internet, large video generation models have demonstrated impressive capabilities that expand the horizons of computer vision and AI [2, 5, 12, 39]. Such models are especially useful in embodied settings, where they can serve as world models, enabling simulation of how the dynamics of the world will evolve given actions [6, 37]. Such an embodied world model can then significantly benefit tasks like path planning and navigation, enabling agents to make decisions based on simulated interactions before acting in the real world and enabling agents to be purely trained on simulated real-world interactions.

However, a fundamental challenge exists for embodied world models: the underlying state of the world is represented as a single image or chunk of images, preventing existing world models from consistently simulating full 3D environments where much of the generated world remain unobserved at any given moment [16]. As a result, models are myopic and often generate new elements that conflict with its historical context—thereby harming the internal consistency required for real-world fidelity. These contradictions undermine the world models' capacity to track evolving environments and plan over extended horizons, ultimately constraining their utility for navigation, manipulation, and other tasks demanding stable, long-term interaction. As illustrated in Figure 1, given a context video defining a 3D scene, the world model without 3D memory fails to retain the unobserved regions once they move out of view. Compared to the ground truth frames, not only do items like the painting and table disappear, but the room structure turns into an entirely different space—showcasing a contradiction with the provided context.

To solve this issue, we present Persistent Embodied World Model, an approach to learning an accurate embodied world model by explicitly incorporating a 3D memory into world models. Real-

---

[*] HKUST   [†] UMass Amherst   [‡] MIT   [§] Independent researcher

39th Conference on Neural Information Processing Systems (NeurIPS 2025).

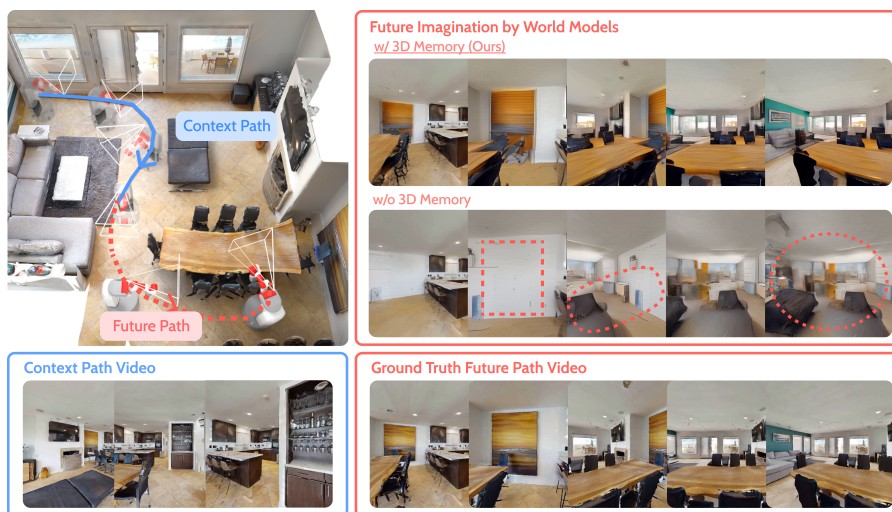

Figure 1: **3D Persistent Video Generation.** Given the context video that defines the top-left 3D scene, the baseline world model deviates from this layout and introduces contradictory elements. In contrast, with 3D memory, our approach preserves observed structures and generates content consistent with the original context.

world scenes naturally unfold in three dimensions, so our model constructs a volumetric memory representation by populating 3D grids with DINO features [22] representing previously generated video frames, thereby capturing the spatial relationships within the environment. To enable accurately capture and generate the 3D geometry in an environment, our model processes and generates RGB-D data to preserve critical geometric cues in the memory's 3D structure. To enable 3D consistent simulation, our model converts an agent's actions into a corresponding relative camera pose change in the map, enabling effective retrieval from the 3D memory and ensuring consistent viewpoints and content across frames. By uniting 3D-structured memory, depth-aware generation, and precise camera control, our model not only predicts future observations under action control but also faithfully reconstructs past scenes with a high degree of spatial coherence. In Figure 1, the 3D memory module allows our method to preserve previously observed 3D structure, maintaining coherent scene content throughout generation. In contrast, a model without the 3D memory progressively deviates from the given context.

Our empirical results show significant enhancement for both the visual quality and consistency of video generation, highlighting the effectiveness of our proposed model and its associated memory mechanisms. In particular, these results confirm that incorporating a 3D memory promotes 3D persistence of embodied video generation, ensuring consistency with both previously generated and observed frames. Additionally, this persistence offers substantial advantages for the downstream robotic applications, including ranking the sampled action trajectories, planning with model predictive control, and policy training in the video simulators.

In summary, our contributions are as follows:

1. We propose the Persistent Embodied World Models, an embodied world model system which incorporates 3D memory into video diffusion models, enabling persistent video generation.

2. We empirically demonstrate that our model offers significant improvement for both the visual quality and consistency of video generation.

3. We show how our model can be used in downstream embodied application such as planning and policy training.

## 2   Method

In this section, we present our proposed method in detail. We first give some formulations about world models with memory and action-driven video generation models in Section 2.1. We then propose our 3D memory approach, which introduces the capability of 3D spatial understanding and

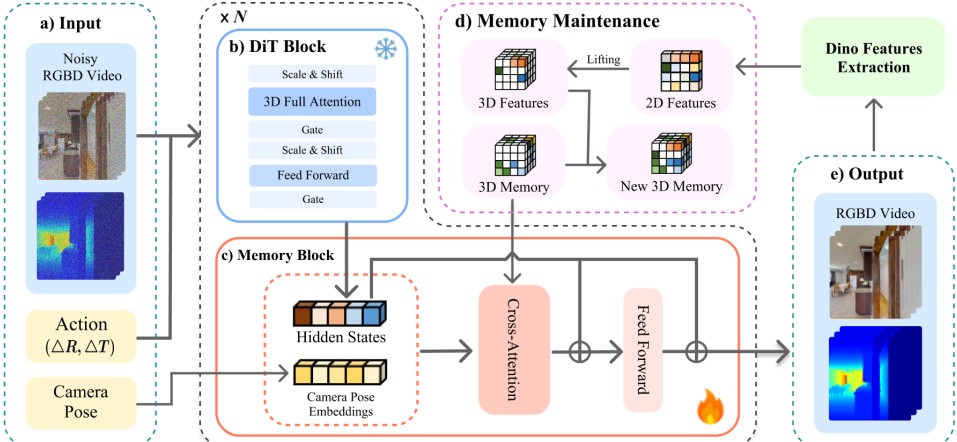

Figure 2: **Overview of our framework.** (a) Our model takes the current RGB-D observation, action, and 3D memory as input and synthesizes an RGB-D video. (d) The memory is incrementally updated after each video generation. (b, c) We train the memory blocks only and freeze DiT blocks of the video models in the second training stage.

3D map features into the video models in Section 2.2. Finally, we describe the training pipeline and potential applications of our model for robotics in Section 2.3.

## 2.1 Formulation

**World Models with Memory**. We formulate our proposed model as action-guided video generation models with a 3D feature map memory. Specifically, our model takes the current image observations $O_t$, agent actions $A_t$ and 3D feature map memory $M$ as input, and then predicts the future observations $\{O_{t+1}, \cdots, O_{t+H}\}$ that the agent will see if it executes the given action $A_t$. In this paper, the observations $O_t$ refer to RGB images and depths. Agent's actions $A_t$ refer to the navigation commands (e.g., move forward/backward, turn left/right) and interaction commands (e.g., pick objects, move objects). The 3D feature map memory $M$ is initialized to all zero values, and then is incrementally updated given the observations of the agent. The memory $M$ enables our model to not only imagine unknown areas but also maintain consistent generations in previously seen areas.

**Action Representation**. In this paper, we convert action commands $A_t$ into a corresponding relative camera pose transformation $C^*$. We then explicitly condition video generation on the computed relative camera pose of each action. Each relative camera pose is represented as the intrinsic matrix $\mathbf{K}$ and extrinsic matrix $\mathbf{E} = [\mathbf{R}|\mathbf{t}]$. However, some works [14] indicate that direct conditioning video generation on the raw camera poses complicates the correlation between these values and image pixels, restricting the ability to control visual details precisely. To more accurately condition on relative camera transform, we use the Plücker embedding [29] to represent each camera transform (more details in Appendix A.2).

**Action-Driven Video Generation**. To leverage the pre-trained generation abilities, we use CogVideoX [39] as the backbone of our model. Previous works [14] mostly utilize cross-attention for the action-driven and camera-control. However, we found that directly concatenating Plücker camera embeddings with input image channels is better for precise pixel-wise control for CogVideoX.

Given a video of shape $T \times H \times W$, 3D VAE of CogVideoX will compress the video and generate the video latent of shape $\frac{T}{q} \times \frac{H}{p} \times \frac{W}{p}$. We also downsample the Plücker camera embeddings with the same compression rate using UnPixelShuffle layers [28]. We concatenate the downsampled camera embeddings with video latent and extend the original patch embeddings to accept new channels.

## 2.2 Video Generation with 3D Map

To enhance the representation of scene information, we create a volumetric memory representation by filling 3D grids with DINO features [22]. Constructing a 3D map from one or several images includes

---

* Parts of an action that do not correspond to a camera pose transformation can be encoded separately

several steps: (1) feature extraction, (2) feature unprojection, (3) aggregation. We first extract image features $F$ via DINO-v2 [22], a pre-trained image encoder, for each image $I$. Then, we lift 2D image feature $F$ into the 3D space. We unproject each pixel to a 3D grid in the world coordinate space using depth, intrinsic and extrinsic matrix. Finally, we aggregate each grid in the DINO-Map through max-pooling. For efficiency, we extract the meaningful grids from the 3D map. To maintain 3D spatial relations, we then concatenate each grid with corresponding 3D sinusoidal absolute position embeddings.

A key challenge in combining video models with the 3D spatial feature map is that most video models are limited in the abilities of 3D spatial understanding. In order to accurately model such relationships, we inject depth information into the video models. We generate RGB-D videos instead of only RGB videos to introduce 3D-aware information into the video latent. With generating and processing depth information, our model can maintain the 3D geometry and spatial information. Drawing from TesserAct [42], we separately encode RGB and depth with 3D VAE. We concatenate RGB and depth and extend the input and output layers to accept and output depth channels.

To inject DINO-Map into the video diffusion models, we design the cross-attention expert block as our memory block. We additionally concatenate camera embeddings $C$ to the video hidden states $H$ such that the model has the information of both camera pose and depth. Thus, the model is able to correlate the hidden states with the corresponding 3D feature grids. Drawing from the CogVideoX, we use expert adaptive layernorms to improve the alignment across two feature spaces. We regress the different scale and shift parameters $\alpha$, $\beta$, and $\gamma$ from the time embeddings $t$ for the video latent and map latent separately. Specifically, we define our memory blocks as:

$$
\begin{aligned}
H_{norm}, M_{norm}, \alpha_H, \alpha_M &= \texttt{norm}_1(H, M, t) \\
H &= H + \alpha_M \texttt{Attn}(H_{norm}, M_{norm}) \\
H &= H + \alpha_H \texttt{ff}(\texttt{norm}_2(H))
\end{aligned}
\tag{1}
$$

where $\texttt{norm}_1$ is expert adaptive layernorm, $\texttt{norm}_2$ is layernorm, $\texttt{Attn}$ is the cross-attention layer and $\texttt{ff}$ is the feed forward layer. As illustrated in Figure 2, we inject the memory blocks after each original transformer block.

## 2.3 Training and Application

**Training**. Training large video diffusion models directly on 3D feature maps is computationally prohibitive. To address this, our approach involves two training stages. We begin by fine-tuning CogvideoX without the memory block on our dataset. The training objective of this stage is:

$$
\mathcal{L} = \mathbb{E}_{z^i, i, \epsilon}[\|\epsilon_\theta(z^i, i | o_t, a_t, c) - \epsilon\|^2]
\tag{2}
$$

where $\epsilon_\theta$ is video models, $z^i$ is video latents, and $o_t$, $a_t$, $c$ represent observation, action and camera pose. We use superscripts $i \in [0, N]$ to denote diffusion steps (for example, $z^i$) and $\epsilon$ as the added noise. The aim of this training stage is threefold: **1)** Adapt pre-trained video models to the specific domain of our dataset. **2)** Integrate action-conditioned controls to enable precise guidance over the generated video. **3)** Optimize video models to generate RGB-D videos, producing both color (RGB) and depth (D) frames simultaneously, which is critical for further map construction.

Next, in the second training stage, we train the memory blocks only and freeze other parameters. The training objective of this stage is formulated as:

$$
\mathcal{L} = \mathbb{E}_{z^i, i, \epsilon}[\|\epsilon_\theta(z^i, i | o_t, a_t, c, M) - \epsilon\|^2]
\tag{3}
$$

where $M$ is the 3D feature map. This process teaches the model to exploit the information in maps but also maintains the generation abilities of the original model.

**Applications in Embodied Domains**. We next describe how to leverage our model in downstream embodied applications such as model predictive control (Algorithm 1 (c)) or policy training. Overall, our model with memory serves as a temporally consistent dynamics model to simulate agent-environment interactions.

To enable an agent to accurately choose actions in an environment, we can combine our model with model predictive control. Given a reward function $R(V)$ that the agent is optimizing, we can optimize an action

$$
a^* = \arg\max_a R(V),
\tag{4}
$$

Table 1: **Video Generation Results**. Our model outperforms baselines in both visual quality and consistency.

| Method | PSNR ↑ | SSIM ↑ | LPIPS ↓ | FVD ↓ | DreamSim ↓ | SRC ↑ |
|---|---|---|---|---|---|---|
| NWM | $17.5_{\pm 0.05}$ | $0.66_{\pm 0.01}$ | $0.31_{\pm 0.01}$ | $194_{\pm 4.3}$ | $0.247_{\pm 0.006}$ | $63.4_{\pm 0.56}$ |
| Image Memory | $19.0_{\pm 0.25}$ | $0.68_{\pm 0.01}$ | $0.27_{\pm 0.01}$ | $124_{\pm 3.7}$ | $0.184_{\pm 0.009}$ | $69.4_{\pm 0.72}$ |
| Ours (w/o depth) | $20.6_{\pm 0.12}$ | $0.67_{\pm 0.01}$ | $0.22_{\pm 0.01}$ | $114_{\pm 2.5}$ | $0.118_{\pm 0.002}$ | $77.8_{\pm 0.24}$ |
| Ours (w/ 2D-Map) | $21.6_{\pm 0.15}$ | $0.75_{\pm 0.01}$ | $0.17_{\pm 0.01}$ | $98_{\pm 2.3}$ | $0.097_{\pm 0.002}$ | $79.2_{\pm 0.22}$ |
| Ours | $\mathbf{22.5}_{\pm 0.05}$ | $\mathbf{0.76}_{\pm 0.01}$ | $\mathbf{0.16}_{\pm 0.01}$ | $\mathbf{92}_{\pm 2.0}$ | $\mathbf{0.086}_{\pm 0.001}$ | $\mathbf{81.7}_{\pm 0.10}$ |

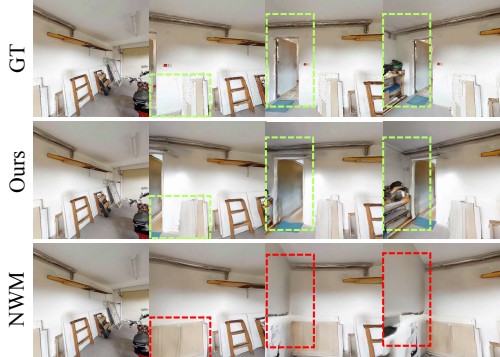 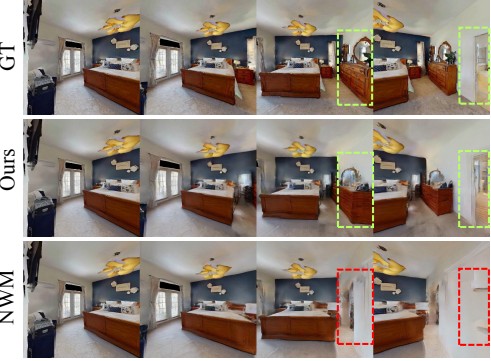

Figure 3: **Qualitative Comparison of Ours with baselines.** Given 3D memory and camera trajectory, the videos generated by our model are high-quality and closely match the ground truth, while NWM [2], without the memory mechanism, generates new contents that conflict with the ground truth. We use green boxes to show consistency and red boxes to show conflict.

which we then execute in the environment. The above optimization objective searches for sequences of actions $a$ so that the image observations in the generated video $V$ from our method maximizes the reward function $r(s)$. When optimizing Equation 4, we can either use sampled action chunks from a pre-trained policy [23] or directly search in the space of the actions using optimization methods such as the cross entropy method [9]. In comparison to directly learning a policy, using MPC to obtain actions enables the agent to more precisely select actions subject to the dynamics of the world, where the addition of memory in our model enables more faithful optimization in partially observable embodied environments.

Alternatively, our model can also be used as a grounded simulator to generate data for policy learning in unseen environments. In this setting, we initialize the map $M$ from a few-shot images of the unseen environment we wish to adapt to. We can then simulate trajectories of interaction in this environment by repeatedly using the policy to generate actions, using our video model to generate future observations given these actions, and then updating the map $M$ with these observed future observations. We can then use hindsight relabeling [36] to label trajectories with goals and rewards, and use the downstream data to train and improve the policy iteratively.

## 3 Experiment

We describe the experimental setting and our design choices and compare our model to previous approaches. We report the results using three random seeds.

### 3.1 Dataset

We collect our dataset in the Habitat Simulation [32] with about 1,000 scenes from HM3D [24]. We split the scenes into training scenes and test scenes. Though the visual quality of Habitat is not perfect, the reason we still choose this is that the scenes from Habitat include multi-rooms and are much larger than the real-world dataset [3, 45]. In addition, Habitat supports the potential to collect data that how the agents interact with the environment. Our dataset includes about 50k trajectories with the most 500 steps. Additional details are included in Appendix A.1.

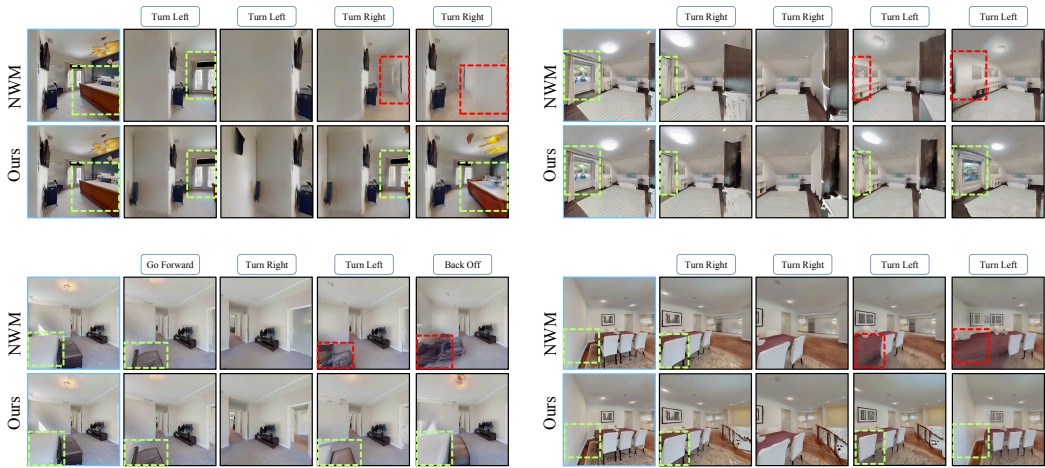

Figure 4: **Qualitative Results of Consistency Generation.** We autoregressively generate videos four times and present the final images from each generation. Our model can generate consistent content after revisiting the same location.

## 3.2 Video Generation

We first evaluate the capability of our model and other baselines to generate temporally coherent and geometrically consistent videos from memory.

**Baselines**. We compare our model with several baselines:

- Navigation World Model (NWM) [2][†]. NMW is a controllable egocentric video generation model. It predicts future observations based on past observations and actions.

- Image Memory. This baseline is drawn from 3D-Mem [38], which is previous used for VQA. It leverages a compact set of informative snapshot images as 3D scene representation. We adopt the snapshot images as the image memory for the video models.

- Ours (w/o depth). This is an ablative baseline trained with RGB videos without depth information.

- Ours (w/ 2D-Map). This is also an ablative baseline trained with 2D feature maps.

**Evaluation**. We evaluate the overall performance of video generate according to FVD [33]. To assess the persistence of the scene information between the videos and memories, we use peak signal-to-noise ratio (PSNR), SSIM [34], LPIPS [41] and DreamSim [11] to measure the frame-wise similarity between the generation and ground truth. We mainly focus on how well our model and other baselines generate coherent and consistent content from memory. Thus, we construct maps based on the previously visited observations as memory. For a fair comparison, all baseline models were configured to have an equal number of parameters. And NWM would take an empty memory as the conditioning input.

**Generation Results**. We first assess the effectiveness of our model. The results are highlighted in Table 1. Our method gains a significant improvement over the baseline without memories (NWM) and surpasses all other baselines with memories across all metrics. Specially, our model reaches a notable FVD score of 92, significantly lower than those of other methods, such as NWM (194) and Image Memory (124). Furthermore, the better PSNR and SSIM scores also reflect that our model excels in visual fidelity. The poor performance of NWM highlights the critical importance of incorporating memory mechanisms, with baselines with an underlying memory mechanism substantially outperforming it. Image Memory and ours (with 2D-Map), which consider 2D representations for the memory, lack 3D spatial awareness, resulting in the degradation of the performance. Comparison with ours (w/o depth) indicates that incorporating depth information enhances the correlation between the hidden states of the video and 3D feature grids. As shown

---

[†] Since the original paper has not released the codes and the architecture of NMW is a conditional diffusion transformer model, similar to our backbone CogvideoX [39]., we implemented it based on our backbone.

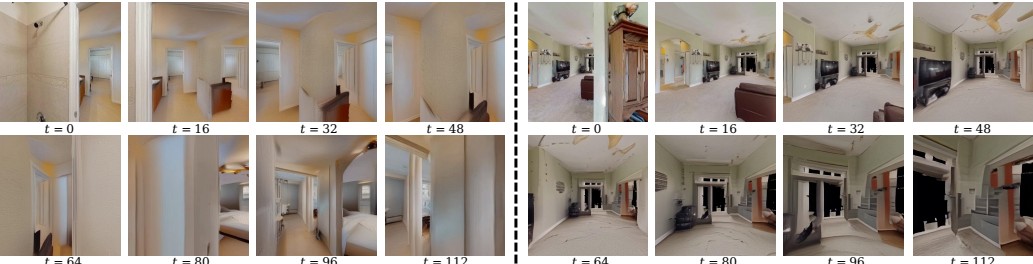

Figure 5: **Qualitative Examples of Long Video Generation.** Our model can generate long videos with memory.

in Figure 3, NWM generates new elements that conflict with the real scene while our model can accurately replicate elements.

**Persistent Generation**. Next, we evaluate models using the metric Scene Revisit Consistency (SRC) from SlowFast [16] to access the consistency of a video observed in the same location when revisited through reverse actions. SRC is determined by calculating the cosine similarities between visual features of the first visit and subsequent visits. We use DINO-v2 [22] to extract the visual features. The results are highlighted in Table 1. We find that baselines with memory mechanisms all have a better performance than NWM, among which our model achieves the best performance. Figure 4 shows the quantitative results of our model and baselines without memory. Our model maintains long consistent generations after turning left twice and right back.

Table 2: **Experimental Results of Ranking Trajectories.** Our model exhibits superior performance compared with baselines.

| Method | ATE $\downarrow$ | RPE $\downarrow$ | SIM $\uparrow$ |
|---|---|---|---|
| NoMaD | $4.94_{\pm 0.2}$ | $4.86_{\pm 0.4}$ | $36.8_{\pm 1.2}$ |
| NWM ($\times 8$) | $4.82_{\pm 0.2}$ | $3.62_{\pm 0.2}$ | $59.7_{\pm 3.5}$ |
| Ours ($\times 8$) | $4.80_{\pm 0.1}$ | $3.58_{\pm 0.1}$ | $62.5_{\pm 3.5}$ |
| NWM ($\times 16$) | $4.54_{\pm 0.1}$ | $3.51_{\pm 0.1}$ | $68.7_{\pm 1.4}$ |
| Ours ($\times 16$) | $\mathbf{4.47}_{\pm 0.1}$ | $\mathbf{3.28}_{\pm 0.1}$ | $\mathbf{70.8}_{\pm 1.7}$ |

### 3.3 Planning with World Models

In this section, we describe how our model can help improve the performance of robotic navigation policies as illustrated in Section 2.3. One major challenge for the previous world model approaches is that they don't have ability to capture the 3D structure of the entire environment, leading to unreliable guidance to the robotic policies. Our model, with 3D persistent memory map, can ground generated videos to the real environment. Similar to NWM [2], we plan with the world models by generating videos of length 8.

**Ranking Trajectories**. We leverage the video models to rank the multiple trajectories generated by an existing policy. We use NoMaD [30], a state-of-the-art diffusion policy for robotic navigation, to sample $N = 8, 16$ action trajectories. We use video models to generate future observations under the action trajectories. We rank trajectories by computing the cost function, which is LPIPS similarity between the goal observation and the last frame of the generated videos.

We use Absolute Trajectory Error (ATE) and Relative Pose Error (RPE) [2, 31] to access the global consistency and local accuracy of the final predicted action trajectories. We also evaluate the similarity (SIM) between the final pose and final rotation after executing the action trajectories and real pose and rotation. As shown in Table 2, our model outperforms navigation policies and the baselines without memory mechanisms, demonstrating the efficacy of our approach.

**Model Predictive Control**. We assess how our model can be used with MPC. For simplicity, we use the discrete actions (e.g., move forward, turn left/right). We initialize the uniform distribution over the action space. We randomly sample $N = 60$ action chunks from the current distributions.

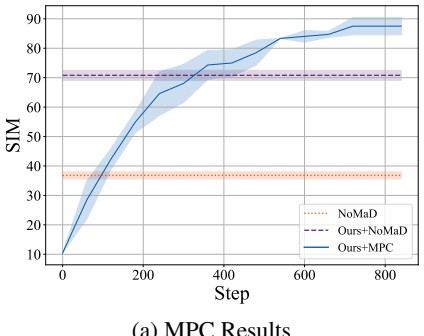

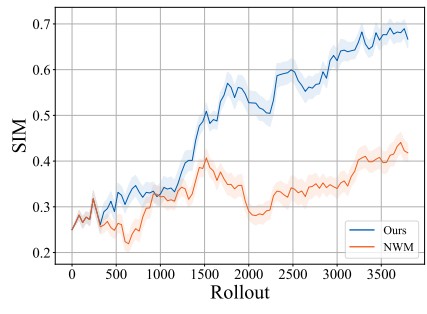

(a) MPC Results          (b) Policy Learning Results

Figure 6: **Experimental Results of MPC and Policy Learning.** **(a)** Through generating more samples, our model with MPC achieves a $17\%$ performance gain compared to ranking action trajectories. **(b)** By leveraging few-shot images stored as memory prior, our model significantly enhances policy learning performance.

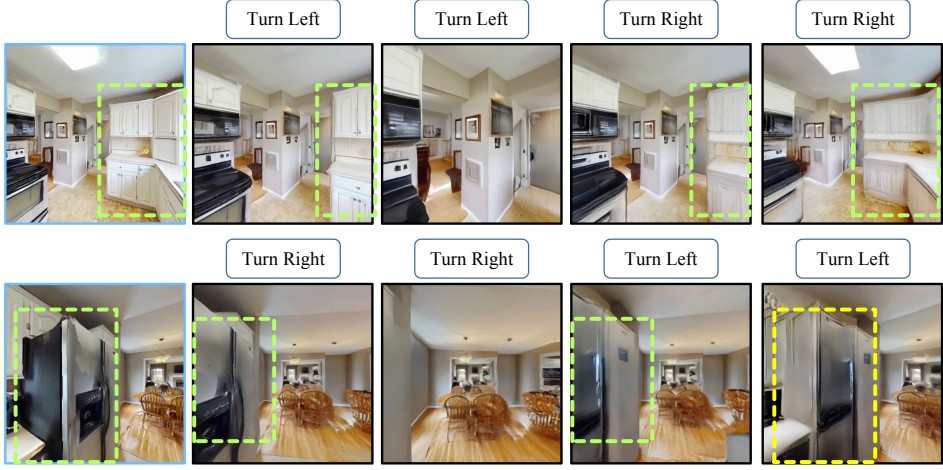

Figure 7: **Qualitative Examples of Unseen Persistent Generation.** Our model can generate persistent videos even in unseen environments.

We generate the potential observations given each action chunk through our model and compute the cost function. We select the top $k = 30\%$ action chunks to update the distributions. The metrics ATE and RPE measure the trajectory-level distance and are not suitable for MPC since two different trajectories can reach a similar final goal. Thus, we only report SIM metrics. The results are shown in Figure 6a. After about 350 iterations, our model with MPC can achieve results similar with ours with NoMad. Our model with MPC achieves competitive results, SIM of $87.5$ after about 720 iterations.

**Policy Learning in New Environment**. As described in Section 2.3, our proposed model has the ability to support the policy training in a new environment with few-shot images. Then, we provide a proof-of-concept by fine-tuning the previous agent policies in the new environments. The results are shown in Figure 6b. Our model can boost policy learning.

## 3.4 Additional Video Generation Results

Finally, we demonstrate the additional capabilities of our model, particularly in generating extended video sequences and generalizing to novel scenes.

**Long Video Generation**. We find that our approach is able to synthesize very long video trajectories while maintaining 3D consistency across extended time horizons by incrementally updating memory. To illustrate this, we generate a total of 112 video frames autoregressively in Figure 5.

**Simulation in Unseen Scenes**. We further illustrate how our approach can generalize well and simulate new unseen scenes. The results are shown in Figure 7, while a modest decline in visual

fidelity occurs compared to training domains, the framework robustly preserves object permanence and scene consistency, underscoring its adaptability to novel scenes.

## 4 Related work

**Embodied World Models**. Significant research efforts have developed generation models as world models in gaming [1, 6], autonomous driving [12, 20], and robotics [4, 8, 23, 36, 43]. Prior works [17, 44] have primarily focused on training models using low-dimensional state and action representations on simulation data. These approaches leverage trajectory generation to enable robot planning. However, such methods face significant challenges in scaling to high-dimensional observation spaces. UniPi [10], addresses this limitation by employing a video diffusion model to directly predict future visual frames subsequently coupled with an inverse dynamics model to infer action. Uni-Sim [36] and Genie [6] demonstrate the utility of world models for training agents by enabling policy learning without direct environmental interaction during training. However, existing world model frameworks in robotics remain largely restricted to simplified settings, such as table-top manipulation tasks, limiting their applicability in real-world scenarios requiring complex spatial reasoning. Some works [18, 19] propose visual world models for the indoor navigation tasks. Most similar to our work, Navigation World Models [2] introduces a controllable video generation framework conditioned on navigation commands to simulate environment dynamics. While promising, this method lacks an explicit memory architecture, leading to scene inconsistency. Our framework integrates a memory mechanism that enables consistent long-horizon imagination, ensuring coherence with previously generated content.

**Video Generation and Memory**. A fundamental challenge in persistent video generation stems from computational memory limitations, which restrict the number of frames processed in a single forward pass. Previous approaches [13, 15, 26, 37] condition each generated video chunk solely on its immediate predecessor, creating truncated context windows that discard prior scene history. This results in temporal fragmentation and inconsistencies when revisiting earlier spatial contexts. Recent work [16], utilizes a temporary LoRA module that embeds the episodic memory in its parameters in video diffusion models. However, it requires additional training time during inference, and its reliance on 2D latent representations neglects 3D spatial priors. Persistent Nature [7] models the 3D world as a 2D terrain and render the video via NeRF [21]. Alternative approaches like InfiniCube [20] leverage structured high-definition (HD) maps to enforce geometric consistency in long videos, but such maps are inherently static, simplistic, and confined to autonomous driving. To overcome these limitations, we propose a 3D feature map that serves as a dynamic memory buffer, explicitly encoding semantic attributes and spatial geometry to maintain coherent environmental representations across extended temporal horizons.

## 5 Conclusion

In conclusion, we present the Persistent Embodied World Model, a novel framework that integrates 3D memory structures into video diffusion models to achieve persistent, spatially coherent video generation. Our model demonstrates significant advancements in both visual fidelity and temporal consistency for generation. Crucially, the system not only synthesizes plausible content for unexplored environments but also maintains geometric and semantic coherence in previously observed scenes. Furthermore, we showcase the practical utility of our model in embodied AI to enhance robotic planning and policy training.

**Limitations and Future Works**. One limitation of our work is that we need the data with depth. Most datasets either don't include depth information [45] or are limited in the diversity of trajectories. [3, 40]. To apply our approach to larger real-world datasets, one direction is to leverage pre-trained depth estimation models (e.g., Depth Anything [35]) to estimate depth in videos. We can further use a mixture of simulation data and real-world data to further improve data diversity.

Additionally, our constructed 3D maps of the environment do not model the dynamic evolution of the environment over time. This prevents our approach from simulating dynamics of many embodied environments where the surroundings in the environment are constantly changing – *i.e.* moving cars in traffic or other inhabitants in a house. Further learning an additional dynamics model on top of the 3D spatial memory to model such changes would be an interesting direction of future work.

## Acknowledgments and Disclosure of Funding

Siyuan Zhou was partially supported by a Research Impact Fund project (RIF R6003-21) and a General Research Fund project (GRF 16203224) funded by the Research Grants Council (RGC) of the Hong Kong Government.

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

# A Experimental Details

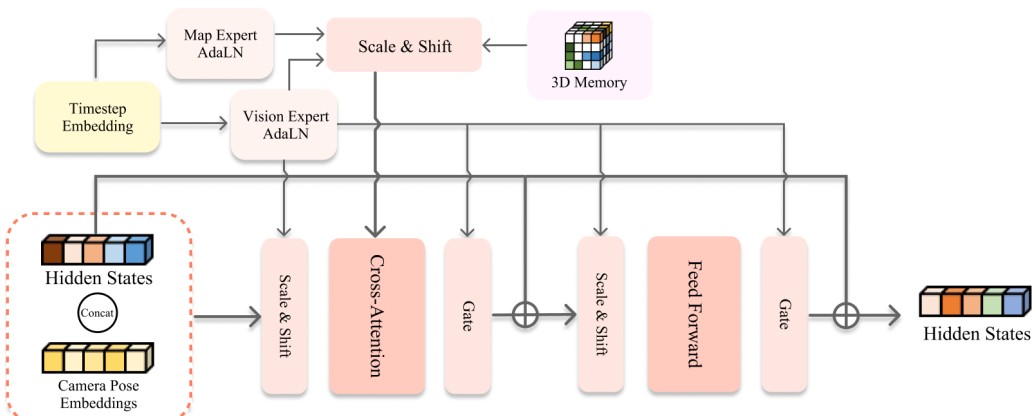

Figure 8: **Detailed Implementation of the Memory Block.**

## A.1 Dataset

We compare different datasets in Table 3. Our dataset collected from HM3D is a large dataset satisfying multi-room and depth support. Though our dataset mainly includes navigation actions, it's feasible to be extended to interaction commands since is collected from the simulation Habitat [32][‡].

We collect our dataset by randomly sampling start and goal positions in the scene and following the shortest path at most 500 steps.

Sometimes, the agent would already have much information about the scenes. Thus, the agent has possibility to turn around to get scene information initially when collecting data.

Table 3: **Comparison with Other Datasets.**

| Dataset | Num. Videos | Room | Depth | Allowed Interaction |
|---|---|---|---|---|
| Ours | $50K$ | Many | ✔ | ✔ |
| RealEstate10K [45] | $10K$ | Many | ✘ | ✘ |
| Scannet++ [40] | $1.8K$ | Many | ✔ | ✘ |
| ARKitScenes [3] | $3.1K$ | Few | ✔ | ✘ |

Table 4: **Details about 3D VAE.**

| | RGB or Depth | Camera |
|---|---|---|
| Input | $9 \times 512 \times 512 \times 3$ | $9 \times 512 \times 512 \times 6$ |
| Compression | 3D VAE | Mean-Pooling & UnPixelShuffle |
| Latent | $3 \times 64 \times 64 \times 16$ | $3 \times 64 \times 64 \times 24$ |

## A.2 Implementation Details

We use CogVideoX [39][§] as our backbone. CogVideoX is a transformer-based architecture. We modify the PatchEmbed Layer to accept the depth latents and camera embeddings. In particular, we expand the input channel of the original convolution network (or MLP in CogVideoX 1.5) and copy the original weights. Expanded parameters are zero initialized. Similarly, we modify the output layer to output the depth latents.

The architecture of our memory block is illustrated in Figure 8. We concatenate video hidden states with camera embedding. Similarly to CogvideoX [39], we normalize the hidden states and 3D

---

[‡] https://github.com/facebookresearch/habitat-lab; MIT license.  [§] https://github.com/THUDM/CogVideo; Apache-2.0 license

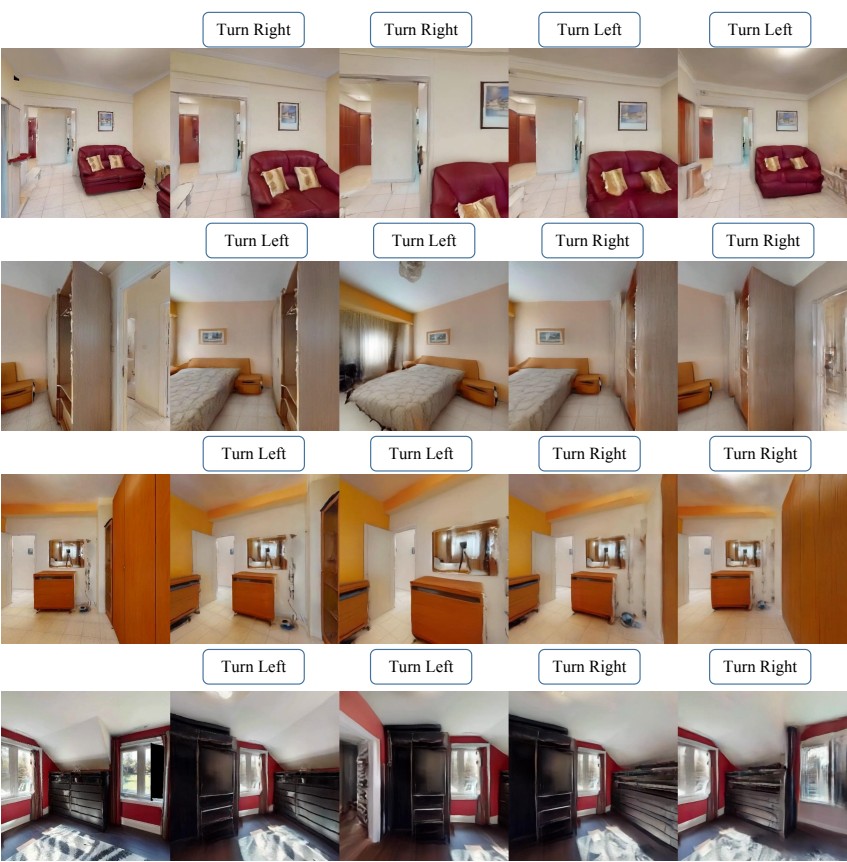

Figure 9: **Consistency Results.**

memory via the expert adaptive layernorm separately. Instead of 3D full attention of CogvideoX, we use Cross-Attention Layer for efficiency. The number of additional parameters for all memory blocks is nearly 1B.

**3D VAE**. We use 3D VAE from CogVideoX to compress RGB-D videos to video latents. We feed RGB and depth separately to 3D VAE, which compresses $9 \times 512 \times 512$ frames to $3 \times 64 \times 64$ latents. We also downsample the plücker embedding with the same compression rate. In particular, we use mean-pooling to downsample the spatial dimension and UnPixelShuffle layers [28] to compress the temporal dimension.

**Memory**. we create a volumetric memory representation by filling 3D grids with DINO features. The shape of 3D grid map is $256 \times 32 \times 256 \times 384$ and the size of each grid is $0.25m \times 1m \times 0.25m$. We first extract image features $F$ via DINO-v2 [22] and upsample the features by bilinear interpolation. Then, we lift 2D image feature $F$ into the 3D space. We unproject each pixel to a 3D grid in the world coordinate space using depth, intrinsic and extrinsic matrix. Finally, we aggregate each grid in the DINO-Map through max-pooling.

Our memory can be easily extended to outdoor environments. We can maintain a large and unbounded grid map and extract the local map that is fed into the video diffusion models based on the camera pose. We leave it for future work.

**Action Representation**. To more accurately condition on relative camera transform, we use the Plücker embedding [29] to represent each camera transform. Specifically, an intrinsic matrix $\mathbf{K}$ and extrinsic matrix $\mathbf{E}$ is encoded as an image $P \in \mathcal{R}^{h \times w \times 6}$. The value of pixel $(u, v)$ in the image is represented as a concatenated tuple $(o \times d, d)$ where $o$ is the ray origin for the pixel and $d_{(u,v)}$ is the ray direction for the pixel. Specifically, $o$ is the camera center in the world coordinate space, which is

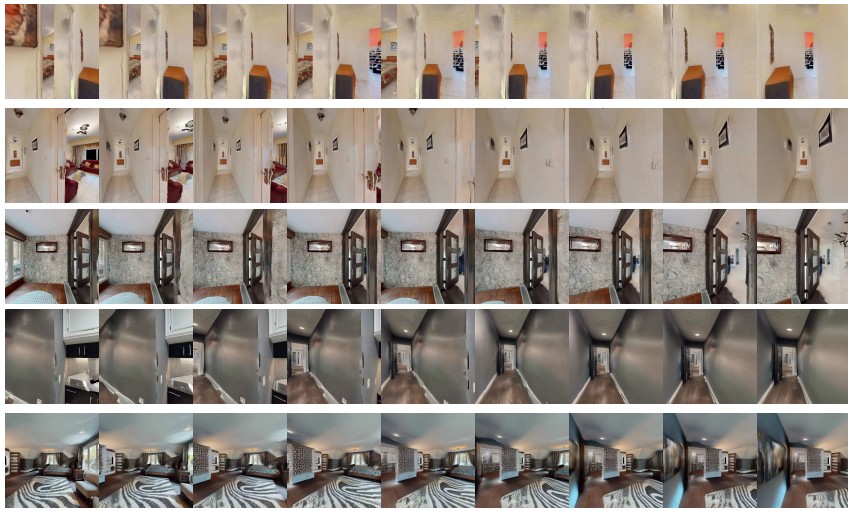

Figure 10: **Generation Results.**

---

**Algorithm 1** Persistent Embodied World Model.

---

**(a) Video Generation**

1: **Definition:** Video Model $p_\theta$, Map $M$, Action Chunk $a_t$, Observation with Depth $o_t$, State $s_t$, Generated Video with Depth $V$, Future Camera Poses $\{c\}$
2: Generate video $V \leftarrow p_\theta(o_t, a_t, M)$
3: Compute camera pose $\{c\}$ through $s_t$ and $a_t$
4: Construct new map $\tilde{M}$ from the generated video $V$ and camera poses $\{c\}$
5: Update map $M$ with $\tilde{M}$

**(b) Construct Map**

1: **Input:** Video $V$, Camera Poses $c$, Image Encoder $E$
2: **Output:** 3D Feature Map $M$
3: RGB $\{I\}$, Depth $\{D\} \leftarrow V$
4: Image feature $F_I \leftarrow E(I)$
5: Compute 3D grid $\{G\}$ via $\{D\}$ and camera poses $c$
6: Compute 3D feature $F$ via $F_I$ and $G$ and concatenate with 3D position embedding $p$
7: Construct Map $M$ by aggregating $F$
8: **Return:** $M$

**(c) Model Predictive Control**

1: **Requirement:** Video Model $p_\theta$, 3D Feature Map $M$, Agent Policies $\pi$, MPC Distributions $\mathbb{P}$
2: Sample action chunks $\{a_t\}$ from the agent policies $\pi$ or the distributions $\mathbb{P}$ from MPC
3: Generate videos $V \leftarrow p_\theta(o_t, a_t, M)$
4: Calculate the cost functions for each sampled video $V$
5: Rank action chunks $\{a_t\}$, update the distributions $\mathbb{P}$ or policies $\pi$

---

$\mathbf{t}$, and $d_{(u,v)}$ is calculated as:

$$d_{(u,v)} = \mathbf{R}\mathbf{K}^{-1}(u\ v\ 1)^T + \mathbf{t} \tag{5}$$

This image representation of actions enables more accurate 3D spatial control from the video model.

### A.3 Training Details

We train our models with frame skip, where training video clips are subsampled by a specific stride. We use various frame stride from 1 to 3 to help the model learn various camera poses. We use AdaM optimizer, with linear warmup and a learning rate of $1e - 4$. Additionally, we utilize bf16 precision for computational efficiency and clip gradients to a maximum norm of 1.0 to stabilize training. We utilized 8 H100 GPUs for training video diffusion models in approximately 3 days. We

adopt v-prediction [27] and use the DDIM sampler [25]. The inference sampling step is set to 50, and the inference time for generating the videos of 9 frames is 5 seconds.

### A.4 Training and Application

We illustrate the process of training and application in Algorithm 1.

## B  Additional Experimental Results

### B.1 Ablation Study on Voxel Size

We evaluated the model with different voxel sizes, as shown in the table 5. Our results indicate that when the voxel size is 256 or larger, model performance remains stable. However, when the voxel size is set below 256, we observe a slight decrease in performance.

Table 5: **Ablation Study on Voxel Size.**

| Voxel Size | LPIPS ↓ | PSNR ↑ | SSIM ↑ |
|---|---|---|---|
| 128 | 0.20 | 20.6 | 0.73 |
| 256 (ours) | 0.16 | 22.5 | 0.76 |
| 384 | 0.16 | 22.2 | 0.77 |

### B.2 Ablation Study on Representations

We evaluate the model with different 3D representation methods, as shown in the table 6. Compared to point cloud representations and the use of CLIP features, our proposed 3D memory approach using DINO features consistently achieves the best performance.

Table 6: **Ablation Study on Representations.**

| Representation | LPIPS ↓ | PSNR ↑ | SSIM ↑ |
|---|---|---|---|
| point cloud | 0.20 | 21.2 | 0.73 |
| clip | 0.18 | 21.5 | 0.75 |
| DINO (ours) | 0.16 | 22.5 | 0.76 |

## C  Additional Qualitative Examples

We show more examples about visual quality and consistency in Figure 9 and  10.

## D  Broader Impacts

This paper introduces research aimed at pushing the boundaries of Machine Learning in the realm of robots. There is a potential risk of collisions with objects and people if the robot system is not properly configured in the real world. Implementing collision detection strategies is a possible approach to mitigate this risk.

