# OpenReview forum: "Learning 3D Persistent Embodied World Models"
_NeurIPS.cc/2025/Conference — NeurIPS 2025 poster_

### Official Review · Reviewer_VcUr · 2025-06-23

**Clarity:** 2
**Significance:** 3
**Originality:** 2
**Rating:** 4
**Confidence:** 3

**Summary:**

The authors suggest that prior work on world models ignore spatial coherency (i.e., room appearance may change upon revisit). In this work, the authors propose a transformer world model that utilizes a 3D map of the environment. Given an RBGD image $o$, action $a$, and memory representation $M$, the authors predict the next observation $o'$. The authors utilize a pretrained image encoder to learn a low dimensional representation of the current image, then use a pretrained transformer backbone to represent the transition function. The authors' major contribution is to model $M$ using a matrix of DiNOv2 features. They take the features from an RGBD image and project these features into the corresponding spatial location in the 3D memory tensor $M$. These features are extracted in the world model via cross attention, enabling better consistency in the latent state representation. Finally, the authors use a diffusion model to generate the next RGBD image.

The authors train their model via a two-step process. First, they finetune their model without $M$. Then, they freeze their model and train $M$. The authors construct a train and test dataset using 50k trajectories collecting from the Habitat simulator. Then, the authors compare to four baselines: two alternate models and two ablations. They ablate their model without depth and with a 2D map. They find that their model outperforms the two alternative models, as well their ablations. Next, the authors compare their world model to two alternative models, demonstrating that their model has better scene revisitation consistency. Finally, the authors evaluate their model for planning purposes in a policy learning and model-predictive control framework. They show that their method produces better policies than two other approaches.

**Questions:**

- Line 41: Accurately -> accurate
- Line 128: "is video models" -> "is the video model"
- Figure 2: What are DiT blocks? These are not defined anywhere in the paper.
Equation 2: What is $\epsilon$? Is this noise? If so, how do you define it? Should it be time-dependent (i.e., $e_i$?)
- Equation 2, 3: What is $c$?
- Equation 4: What is $R(V)$? You define the reward function as $r$.
    - In the following paragraph, you say "The above optimization objective searches for sequences of actions a", but in the preceding paragraph, you find $a^*$ by "optimiz[ing] an action"
- Line 133: You say you "train the memory blocks only and freeze other parameters". The memory blocks themselves are just tensors, not parameters. Which weights do you actually train here? It is just the cross attention weights? Do you train the feedforward network as well?
- Line 178: You perform an ablation of your model without depth? How is a depth-free variant of your model expected to work, given that you use the depth information to project the features into the 3D memory tensor?

**Ethical Concerns:**

["NO or VERY MINOR ethics concerns only"]

**Final Justification:**

I am in-between a borderline reject and accept. Since the authors shared their code with the AC, I will give the authors the benefit of the doubt and vote for borderline accept.

**Limitations:**

Yes

**Paper Formatting Concerns:**

None found.

**Quality:**

2

**Strengths And Weaknesses:**

### Strengths
- The writing style is generally good
- The authors provide many useful figures comparing their approaches to others
- Their topic, consistency in world models, is an important and timely topic

### Weaknesses
- Given the focus on video generation, it is suspicious that the authors do not provide any generated video. It is easy to cherry-pick a couple nice looking frames for figures, while having poor video trajectories. This alone is grounds for rejection, in my opinion.
- The authors do not provide any code to validate their approach or compare the results to other baselines. Again, another reason for rejection. Given the recent explosion on the number of world model papers published, reproducability is a necessity.
- Almost every equation in the paper confused me. The authors are imprecise with their equations and definitions, and it greatly hurts understanding. See the *Questions* section for more information.
- The "no-depth" ablation does not make sense, please see *Questions* for more information.

> However, a fundamental challenge exists for embodied world models: the underlying state of the world is represented as a single image or chunk of images

This is false. The paper that invented the term "world model" represents the state using an RNN [1]. A world model, by definition, relies on the POMDP framework and represents the  latent state using either an RNN or transformer.

---

> ### Author Rebuttal · Authors · 2025-07-31
>
> Dear Reviewer,
>
> We thank the reviewer for the detailed comments and questions. We are glad to hear that you find our paper is well-written and our problem is significant. Here is the response to your question.
>
> > "the underlying state of the world is represented as a single image or chunk of images." ... The paper that invented the term "world model" represents the state using an RNN.
>
> We would like to clarify that in this context, our use of "embodied world models" specifically refers to recent large video diffusion models, as discussed in [1, 2]. We acknowledge that the original "world model" framework employed RNNs for state representation, and we will make this distinction clearer in our revision.
>
> [1] Bar, Amir, et al. "Navigation world models." Proceedings of the Computer Vision and Pattern Recognition Conference. 2025.
>
> [2] Yang, Mengjiao, et al. "Learning interactive real-world simulators." arXiv preprint arXiv:2310.06114 1.2 (2023): 6.
> > do not provide any generated video.
>
> In the main paper and Appendix, we included detailed frame-by-frame visualizations to highlight both the strengths and potential failure cases that could be missed in video playback. Unfortunately, due to rebuttal submission policies, we are unable to upload or share additional video material during this stage.
>
>
> > do not provide any code
>
> As stated in our submission, we will open-source both our code and dataset upon acceptance of the paper. We would send the anonymous link related to the code if available at this stage.
>
>
> > What are DiT blocks?
>
> DiT blocks [1] refer to using Transformers as the backbone of diffusion models. In our work, these are implemented in the backbone CogVideoX [2]. We will clarify this in the revision to make it explicit for readers.
>
> [1] Peebles, William, and Saining Xie. "Scalable diffusion models with transformers." Proceedings of the IEEE/CVF international conference on computer vision. 2023.
>
> [2] Yang Z, Teng J, Zheng W, et al. Cogvideox: Text-to-video diffusion models with an expert transformer[J]. arXiv preprint arXiv:2408.06072, 2024.
>
> > Equation 2: What is $\epsilon$? Is this noise? If so, how do you define it? Should it be time-dependent?
>
> $\epsilon$ is noise and is sampled from a Gaussian distribution. This formulation is standard in diffusion models, as also used in CogVideoX [1] and Stable Video Diffusion [2]. We will clarify this notation and provide additional details in the revision.
>
> [1] Yang Z, Teng J, Zheng W, et al. Cogvideox: Text-to-video diffusion models with an expert transformer[J]. arXiv preprint arXiv:2408.06072, 2024.
>
> [2] Yao C H, Xie Y, Voleti V, et al. Sv4d 2.0: Enhancing spatio-temporal consistency in multi-view video diffusion for high-quality 4d generation[J]. arXiv preprint arXiv:2503.16396, 2025.
>
> > Equation 2, 3: What is $c$?
>
> $c$ refers to the camera pose transform, which is defined in Section 2.1.
>
> > Equation 4: What is $R(v)$? You define the reward function as $r$.
>
> Reward function $r$ is the reward of a single step. Return $R(V)$ is the sum of the rewards in the videos. In our paper, $R(V)$ is defined as the reward of the last frame, following Navigation World Model [1].
>
> We will clarify these definitions in the revision. Additionally, we note that it is common in reinforcement learning literature (e.g., Policy Gradient [2]) to denote the return or expected cumulative reward as $R$.
>
> [1] Bar A, Zhou G, Tran D, et al. Navigation world models[C]//Proceedings of the Computer Vision and Pattern Recognition Conference. 2025: 15791-15801.
>
> [2] Sutton R S, McAllester D, Singh S, et al. Policy gradient methods for reinforcement learning with function approximation[J]. Advances in neural information processing systems, 1999, 12.
>
>  > "The above optimization objective searches for sequences of actions a", but in the preceding paragraph, you find $a^*$ by "optimiz[ing] an action"
>
> Model predictive control (MPC) will first initialize the solution space with a Gaussian distribution, then search candidate solutions in the solution space, which will be used to optimize the solution space based on the expected rewards $R(V)$.
>
> > Line 133: You say you "train the memory blocks only and freeze other parameters". The memory blocks themselves are just tensors, not parameters. Which weights do you actually train here? It is just the cross attention weights? Do you train the feedforward network as well?
>
> To be precise, the memory features are tensors, whereas the memory blocks are neural networks that include both cross-attention and feedforward networks, as shown in Figure 2. When we state that we "train the memory blocks only and freeze other parameters," we mean that we update the weights of the cross-attention and feedforward networks within the memory blocks, while keeping the rest of the model parameters fixed. We will clarify this in the revision.
>
>
> > How is a depth-free variant of your model expected to work?
>
> We used pre-trained depth estimation models (Depth Anything v2 [1]) to estimate the depth of the generated videos, and then updated the 3D memory.
>
> [1] Yang, Lihe, et al. "Depth anything v2." Advances in Neural Information Processing Systems 37 (2024): 21875-21911.

---

> > ### Comment · Reviewer_VcUr · 2025-08-02
> >
> > Thank you for the response. Given the recent conference restrictions, I cannot ask for video demos. I will no longer consider the absence of videos in my review.
> >
> > But can you help me understand why you would not provide your code upon submission, and only release the code after acceptance? Reproducability is a big problem in deep learning, so ensuring the paper can be produced and that the ablations/comparisons are fair is very important to me. I audit the code for all the papers I review, and I think this is a necessary task as a reviewer.
> >
> > While I understand diffusion models, it is probably quite clear from my comments that they are not my main area of focus. Formally defining terms like $\epsilon, R$, etc that you consider self-evident is beneficial to others who read your work outside of this subdiscipline. I thank you for your changes.
> >
> > > We used pre-trained depth estimation models (Depth Anything v2 [1]) to estimate the depth of the generated videos, and then updated the 3D memory.
> >
> > Then the following text seems quite misleading
> > > Comparison with ours (w/o depth) indicates that incorporating depth information enhances the correlation between the hidden states of the video and 3D feature grids.
> >
> > So you still incorporate depth information, but
> > - `w/o depth` uses depth information predicted using a neural network
> > - `with depth` uses depth information predicted using the algorithms on your depth sensor
> >
> > What is your w/o depth ablation study actually measuring, then?

---

> > > ### Author Response · Authors · 2025-08-02
> > >
> > > Dear Reviewer,
> > >
> > > Thanks for your detailed feedback.
> > >
> > > > code
> > >
> > > We have already provided the code to the AC who is able to access it.
> > >
> > > > Formally defining terms
> > >
> > > Thanks for your suggestions. We will formally define all terms in the revision, making the paper easier to follow.
> > >
> > > > depth information
> > >
> > > $\textcolor{brown}{\text{w/o depth}}$ doesn't incorporate depth information into the model. It only uses estimated depth information to construct and update the 3D memory.
> > >
> > > This comparison is conducted to verify that, without depth information, video generation models cannot effectively learn information from 3D memory due to their lack of 3D spatial understanding. In order to accurately model such relationships, we inject depth information into the video models.

---

> > > > ### Comment · Reviewer_VcUr · 2025-08-07
> > > >
> > > > We thank you for providing code to the AC and answering our questions. We stress that the authors make use of the supplementary material next time to avoid any reproducibility concerns. We will update our score accordingly.

---

> ### Author Response · Authors · 2025-08-07
>
> Dear Reviewer,
>
> We sincerely appreciate your thoughtful feedback. Your insights and suggestions have been invaluable in guiding our revisions. We will carefully incorporate the points discussed into the final version to enhance its quality.
>
> Thank you once again for your time and consideration.
>
> Best,
>
> The Authors

---

### Official Review · Reviewer_MRYi · 2025-06-26

**Clarity:** 2
**Significance:** 3
**Originality:** 3
**Rating:** 3
**Confidence:** 3

**Summary:**

This paper addresses action-guided video generation from RGB-D inputs: given past observations and control signal, the goal is to synthesize future frames that are consistent with the observed world. Recently, such models are termed 'world models' in the literature (i.e., given past observations and control signal, the task is to predict unobserved video frames based on the internal representation of the world).

This paper's main contribution is persistent, spatial volumetric memory, built by storing visual (DINO) features extracted from newly synthesized frames in a voxel grid. Such construction is possible when input frames consist of image and depth pairs (in contrast to video/camera only inputs).

To generate output images (and corresponding depth maps), the proposed model encodes video + depth, combines encoded RGBD input with relative poses (wrt. camera pose) using Pluecker embeddings, and uses cross-attention layer that augments encoded representation with 3D spatial features, stored in the spatial memory.

The method is trained and evaluated on the Habitat Simulation dataset using synthetic data (and pixel-perfect depth maps).

**Questions:**

Papers states in 2.2: "For efficiency, we extract the meaningful grids from the 3D map.". Would you please explain?

Also, could authors elaborate on the statement: "A key challenge in combining video models with the 3D spatial feature map is that most video models are limited in the abilities of 3D spatial understanding." To the best of my knowledge, this statement is inconsistent with findings in the literature.


References:
[A] Agarwal, Niket, et al. "Cosmos world foundation model platform for physical ai." arXiv preprint arXiv:2501.03575 (2025).

**Ethical Concerns:**

["NO or VERY MINOR ethics concerns only"]

**Final Justification:**

After the rebuttal, I am getting the sense that the authors did not take our recommendations on how to improve this paper seriously, and I don't think the paper is ready to be accepted in the current form. For example:

Rebuttal "*We respectfully disagree with the notion that our method can only be trained on synthetic data. Recent advances in 3D feedforward models [1, 2] have demonstrated the ability to generate high-quality depth maps from multiple views in the real scenario.*"

This is a claim that is not supported empirically. Either show that this method can be trained on real data, or acknowledge that this is a limitation. The table referenced above still applies for synthetic data only (correct me if I am wrong).

Rebuttal: "*We acknowledge that models such as Cosmos demonstrate strong 3D spatial reasoning; however, these models are trained with extremely large-scale datasets and computational resources, typically accessible only to major companies. In contrast, our approach aims to enhance 3D spatial understanding without requiring massive data or GPU resources.*"

Yes, indeed, but this is not the claim that the paper makes. If this is the claim, it should be stated precisely like this. This is not what the paper states: *A key challenge in combining video models with the 3D spatial feature map is that most video models are limited in the abilities of 3D spatial understanding.* (also, is this really the key challenge in "in combining video models with the 3D spatial feature map")?

Statements in the paper raised above are **not** sufficient reason to recommend rejecting this paper. Still, the paper is full of such issues, and these do compound (also, see reviewer's **VcUr** comments: "*Almost every equation in the paper confused me. The authors are imprecise with their equations and definitions, and it greatly hurts understanding. See the Questions section for more information.*", -- I strongly agree).


At the same time, I do not agree with **VcUr** that the fact that the code wasn't released (or videos) is a sufficient basis for rejection. If so, we would need to hold all papers to the same standards, and these standards should have been outlined in the reviewer guidelines.

I am not strongly objecting to the acceptance of this paper, but given several presentation issues and a lack of trust that these will be addressed, I think this paper should undergo another round of revision.

**Limitations:**

The paper points out that key limitations are the static world assumption (present in the dataset), and assuming perfect depth maps as input.

**Paper Formatting Concerns:**

No concerns regarding formatting.

**Quality:**

3

**Strengths And Weaknesses:**

### Strengths

* The paper is reasonably well written
* The problem statement is crisp and clear: Does maintaining persistent spatial memory and conditioning video generation on this memory improve action-conditioned video generation?
* Proposed approach with spatial memory presents improvements wrt. variety of metrics that asses visual generation quality, and is shown to work exceptionally well for long-term video generation, which is intriguing
* I also appreciate experiments on using the trained model in robotic navigation and planning (solid improvements wrt. ATE and RPE).
* Overall, experimental validation is extensive, covering video generation quality, ranking of trajectories generated by a pre-defined policy, model-predictive control and policy learning. Paper even provides visuals, showcasing long-term video generation, and generated views that were not covered by the camera trajectory (unseen scenes).

### Weaknesses

**Presentation**:
* Fig 2 stages that "Our model takes the current RGB-D observation, action, and 3D memory as input and synthesizes an RGB-D video", but I understand that memory is part, not input, to the model; it is built implicitly. This presentation is confusing.

* The paper often uses references with absolutely no context, making the paper not self-contained. For example: "Drawing from TesserAct [40], we separately encode RGB and depth with 3D VAE."; a better statement would be "we do XYZ, following [40]." Another example of weird sentence without context is "We also downsample the Plücker camera embeddings with the same compression rate using UnPixelShuffle layers [26]." There are many such cases.

**General:**

The proposed work can only be trained on synthetic data. The paper argues that this issue could be mitigated with depth generation models, but I am skeptical. The accumulation of multiple views with depth generation models in a voxelgrid will usually not lead to consistent results; it might be that the proposed work only works with perfect depth data.

I would have hoped to see this method trained and evaluated on at least more challenging synthetic datasets (For example, Fabri et al., MOTSynth, ICCV'21, Virtual KITTI; these could be assessed for video generation without control). I know further work may be needed to handle such dynamic scenes, but it is also great to learn where/how the proposed methodology breaks down.

Nevertheless, I believe that the proposed contribution (and its significance wrt. geometry-consistent RGBD generation) was clearly established on at least one dataset. I think this idea should be discussed in the community and further explored in future work.


I am not very familiar with the state of research in this field, so for the time being, I assume that the selection of dataset and baselines for the evaluation is reasonable. I will revisit this / check again when I see comments of other reviews that may be more familiar with action-conditioned video generation and state-of-the-art in this field.

---

> ### Author Rebuttal · Authors · 2025-07-31
>
> Dear Reviewer,
>
> We appreciate the reviewer for the detailed comments and insightful suggestions. We are glad to hear that you find our paper is well motivated and our experimental validation is extensive. Here is the response to your question.
>
> > "Our model takes the current RGB-D observation, action, and 3D memory as input and synthesizes an RGB-D video", but I understand that memory is part, not input, to the model; it is built implicitly.
>
> We would like to clarify that, in our method, 3D memory is explicitly constructed and integrated into the model by learning memory blocks, as illustrated in Figure 2 and Figure 8 (Appendix). Our memory blocks are composed of expert adaptive layernorms, cross-attention blocks, and feedforward blocks. We will revise the paper to better clarify it.
>
> > The paper often uses references with absolutely no context, making the paper not self-contained. For example: "Drawing from TesserAct [40], we separately encode RGB and depth with 3D VAE."; a better statement would be "we do XYZ, following [40]." Another example of weird sentence without context is "We also downsample the Plücker camera embeddings with the same compression rate using UnPixelShuffle layers [26]." There are many such cases.
>
> We appreciate your suggestions and will thoroughly review all references to ensure they are introduced with appropriate context, making the paper easier to follow.
>
> > The proposed work can only be trained on synthetic data. The paper argues that this issue could be mitigated with depth generation models, but I am skeptical.
>
> > I would have hoped to see this method trained and evaluated on at least more challenging synthetic datasets (For example, Fabri et al., MOTSynth, ICCV'21, Virtual KITTI; these could be assessed for video generation without control).
>
> Thank you for your insightful feedback. We respectfully disagree with the notion that our method can only be trained on synthetic data. Recent advances in 3D feedforward models [1, 2] have demonstrated the ability to generate high-quality depth maps from multiple views in the real scenario.
>
> Despite the added challenges posed by dynamic scenes and noisy depth maps, our memory approach still demonstrates improvements in visual fidelity. These results suggest that our method is robust to moderate noise in the depth input, and we believe that further advancements in depth estimation will enhance the generality and applicability of our approach.
>
> | Virtual KITTI | LPIPS $\downarrow$ | PSNR $\uparrow$ | SSIM $\uparrow$ |
> | -------- | -------- | -------- | -------- |
> | NWM | 0.1989 | 20.2 | 0.625 |
> | Ours | 0.1731 | 21.4 | 0.660 |
>
> [1] Wang, Shuzhe, et al. "Dust3r: Geometric 3d vision made easy." Proceedings of the IEEE/CVF Conference on Computer Vision and Pattern Recognition. 2024.
>
> [2] Wang, Jianyuan, et al. "Vggt: Visual geometry grounded transformer." Proceedings of the Computer Vision and Pattern Recognition Conference. 2025.
>
>
> > Papers states in 2.2: "For efficiency, we extract the meaningful grids from the 3D map.". Would you please explain?
>
> When constructing the 3D memory, some grids may be empty or contain very few points. For efficiency, we filter out these empty or sparsely populated grids and only retain grids with meaningful information. This reduces computation and memory usage without compromising the quality of the 3D memory representation.
>
> > Also, could authors elaborate on the statement: "A key challenge in combining video models with the 3D spatial feature map is that most video models are limited in the abilities of 3D spatial understanding." To the best of my knowledge, this statement is inconsistent with findings in the literature.
>
>
> We acknowledge that models such as Cosmos demonstrate strong 3D spatial reasoning; however, these models are trained with extremely large-scale datasets and computational resources, typically accessible only to major companies. In contrast, our approach aims to enhance 3D spatial understanding without requiring massive data or GPU resources. By leveraging 3D-structured memory, depth-aware generation, and precise camera control, our method enables effective 3D reasoning in a more accessible and resource-efficient framework.

---

> ### Author Response · Authors · 2025-08-07
>
> Dear Reviewer,
>
> > Either show that this method can be trained on real data, or acknowledge that this is a limitation. The table referenced above still applies for synthetic data only
>
> We would like to clarify that the results on Virtual Kitti, while still based on synthetic data, hold meaningful insights. They demonstrate that our model maintains robustness even with noisy depth inputs, which is a step toward addressing the challenges of real-world data where depth estimation is often imperfect. This robustness, we believe, provides valuable evidence for the potential of our approach as depth estimation techniques continue to advance.
>
> Our current experiments are indeed conducted on synthetic data and we have explicitly flagged this limitation in the paper "we need the data with depth". We will strengthen this acknowledgment in the revision to leave no room for ambiguity.
>
> > most video models are limited in the abilities of 3D spatial understanding
>
> We apologize for the imprecision in our earlier statement. While Cosmos is a promising model, it has significant limitations when it comes to 3D spatial understanding. For instance, even Cosmos itself acknowledges it’s still in the early stages and lacks object permanence.
>
> Our key point, supported by our empirical results, is that explicitly incorporating depth information and 3D memory enhances 3D persistence. As shown in Table 1, our model with depth (Ours) outperforms the ablative version without depth (Ours (w/o depth)) across metrics like PSNR, SSIM, and SRC, directly demonstrating that depth-aware generation enhances the correlation between the hidden states of the video and 3D memory.
>
>
> > Almost every equation in the paper confused me
>
>
> Because of length limitations, we didn’t include preliminary background in the paper. We recognize that we assumed that readers already have foundational knowledge in this area.
> However, our equations adhere to standard formulations widely used in diffusion model literature [1, 2]—this is not arbitrary, but grounded in established conventions.
>
> In the revised version, we will significantly expand the preliminary section to include detailed explanations of the diffusion model fundamentals that underpin our equations.
>
> [1] Yang Z, Teng J, Zheng W, et al. Cogvideox: Text-to-video diffusion models with an expert transformer[J]. arXiv preprint arXiv:2408.06072, 2024.
>
> [2] Yao C H, Xie Y, Voleti V, et al. Sv4d 2.0: Enhancing spatio-temporal consistency in multi-view video diffusion for high-quality 4d generation[J]. arXiv preprint arXiv:2503.16396, 2025.

---

### Official Review · Reviewer_VbjC · 2025-07-03

**Clarity:** 3
**Significance:** 3
**Originality:** 3
**Rating:** 5
**Confidence:** 3

**Summary:**

- The authors incorporate 3D memory features into video diffusion models to enhance agent's understanding of the environment. The goal is to build an embodied agent with long horizon and spatial consistency even during occlusions or revisits.
- Instead of direct 3D supervision, the model extracted dino features per image and projected them into a 3D volumetric representation. The output predictions are in RGBD format with built in depth understanding, making the model more robust than 2D approaches.
- The results section presented a clear improvement of visual quality and consistency. The paper also empirically demonstrated that the memory aware model benefits robotics path planning.

**Questions:**

- What are the rationales in choosing the proposed 3D representation method, as opposed to alternatives like point cloud, nerf, InfiniCube? Are there ablation studies run on alternative 3D representations, or ablation studies on not using dino features?
- What’s the long horizon boundary of the given method? While the paper qualitatively illustrates good video generation results up to t=112, at what point does the memory module start to fail?
- What is the end-to-end inference latency for future frame generation and dynamic memory updates, and how does the latency compare with other methods in the ablation? Is the inference time real-time?

**Ethical Concerns:**

["NO or VERY MINOR ethics concerns only"]

**Limitations:**

yes

**Quality:**

3

**Strengths And Weaknesses:**

Strength
- The paper is well motivated to incorporate persistent 3D understanding in the video diffusion model and embodied agent path planning. While there is existing literature on 3D aware video generation, the specific implementation details (e.g. using DINO feature, training memory modules with freezed DiT backbones) provide insights for future papers.
- The paper is well-written, clear and provides solid evidence to support the claims.

Weakness
- The paper lacks discussion on the method scalability. The volumetric representation could be intractable in large and complex scenes, limiting the usage of this representation in real scenarios.
- The paper would benefit from further discussion and experimentation on the specific 3D representation methodology. While briefly mentioned in the related works section, it is unclear how the proposed methodology is superior to alternatives like Nerf and InfiniCube. There is also a lack of ablation study to justify the choice of dino feature, as opposed to alternatives such as using point cloud.
- Like the authors stated in the limitations section, the current setup is constrained, and does not apply to dynamic and changing scenes.

---

> ### Author Rebuttal · Authors · 2025-07-31
>
> Dear Reviewer,
>
> We thank the reviewer for the detailed reviews and insightful suggestions. We are glad to hear that you find our paper is well motivated and well-written. Here is the response to your question.
>
> > The volumetric representation could be intractable in large and complex scenes, limiting the usage of this representation in real scenarios.
>
> We acknowledge the potential scalability challenges of volumetric representations in large and complex environments. However, our video dataset is based on HM3D [1], which contains large-scale 3D reconstructions of diverse real-world locations. Each HM3D scene comprises multiple rooms and a variety of objects, with the largest scene covering an area of 2,172 m². Our approach has demonstrated effective scaling to these large and complex environments. We believe this indicates the practical applicability of our method to real-world scenarios, though we agree that further work on memory efficiency and scalability could further enhance its usability in even larger-scale settings.
>
> [1] Ramakrishnan, Santhosh K., et al. "Habitat-matterport 3d dataset (hm3d): 1000 large-scale 3d environments for embodied ai." arXiv preprint arXiv:2109.08238 (2021).
>
>
> > What are the rationales in choosing the proposed 3D representation method, as opposed to alternatives like point cloud, nerf, InfiniCube? Are there ablation studies run on alternative 3D representations, or ablation studies on not using dino features?
>
> Thanks for the suggestions. We evaluate the model with different 3D representation methods as shown in the table. We experimented with integrating NeRF into the video generation pipeline; however, NeRF requires several minutes to reconstruct a scene, making it impractical for use during training. InfiniCube, on the other hand, relies on structured high-definition maps that are primarily tailored for autonomous driving scenarios, which limits its applicability to our more general settings. Compared to point cloud representations and the use of CLIP features, our proposed 3D memory approach using DINO features consistently achieves the best performance.
>
> | Representation | LPIPS $\downarrow$ | PSNR $\uparrow$ | SSIM $\uparrow$ |
> | -------- | -------- | -------- | -------- |
> | point cloud     |  0.20    | 21.2 | 0.73 |
> | clip | 0.18 |  21.5 | 0.75 |
> | DINO (ours) | 0.16 | 22.5 | 0.76 |
>
>
> > What’s the long horizon boundary of the given method?
>
> The long-horizon capability of our method depends on the complexity and scale of the scene. There is an increasing probability that the memory module may struggle to maintain temporal coherence, leading to degradation in generation quality when $t$ is larger than 112.
>
> > What is the end-to-end inference latency for future frame generation and dynamic memory updates, and how does the latency compare with other methods in the ablation? Is the inference time real-time?
>
> We show the inference latency in the table. At present, our video generation model does not achieve real-time performance. However, we believe that real-time inference is feasible by incorporating recent acceleration techniques such as diffusion-forcing [1, 2]. We plan to explore these optimizations in future work to further reduce latency and enable real-time applications.
>
> |  | generation (second) | memory updates (second) |
> | -------- | -------- | -------- |
> | No Memory     |   7   | 0 |
> | 2d Memory | 10 | <1 |
> | Ours | 10 | <1 |
>
>
> [1] Yin, Tianwei, et al. "From slow bidirectional to fast autoregressive video diffusion models." Proceedings of the Computer Vision and Pattern Recognition Conference. 2025.
>
> [2] Chen, Boyuan, et al. "Diffusion forcing: Next-token prediction meets full-sequence diffusion." Advances in Neural Information Processing Systems 37 (2024): 24081-24125.

---

> ### Author Response · Authors · 2025-08-08
>
> Dear Reviewer,
>
> Thank you sincerely for your thoughtful feedback and for recognizing the significance of our work. We are eager to implement these revisions that address your concerns comprehensively in the final version. Thank you again for your invaluable input.
>
> Best,
>
> The Authors

---

### Official Review · Reviewer_jqaK · 2025-07-03

**Clarity:** 3
**Significance:** 3
**Originality:** 3
**Rating:** 4
**Confidence:** 4

**Summary:**

This paper introduces a persistent embodied world model --  a new method for keeping spatial memory of a room baked into the world model. The authors propose incorporating an explicit 3D memory mechanism into video diffusion models, enabling consistent long-horizon simulation of embodied agents navigating complex environments. The key innovation is maintaining a persistent 3D feature map that gets updated as the agent explores. The model generates RGB-D (color + depth) videos conditioned on current observations, actions, and this 3D memory map. This allows the model to accurately reconstruct previously seen areas when revisiting them, rather than hallucinating new content.

**Questions:**

What type of information is stored in this model volume? Can you linearly decode a semantic map of the room?

What is the sensitivity to voxel size ? How bit of a maximum scene can this approach handle?

How does this compare to existing novel view synthesis and world models out there such as stable video camera? Does the explicit memory approach only show benefits when the trajectories are very long? And in that case how does it compare to simply using an every updating NERF to store a representation of the environment?

**Ethical Concerns:**

["NO or VERY MINOR ethics concerns only"]

**Final Justification:**

I will keep my score as a borderline accept. Though this still means I recommend acceptance (as the name implies). My main concerns with the work are that the evaluations do not exactly match the types of navigation and embodiment tasks that this method would be most useful for, and including them would significantly improve the cause

**Quality:**

3

**Strengths And Weaknesses:**

Strengths:
- The paper proposes a novel 3D feature map integrated as conditioning with video diffusion models to act as spatial memory.
- The model shows significant improvements across multiple metrics and demonstrates practical benefits in downstream tasks like trajectory ranking and model predictive control.
- The paper presents a complete pipeline

Weakness:
- There could be some more evaluations relating the quality of the generations against Novel view synthesis baselines such as Stable Virtual Camera
- There could be some evaluations of the physical collisions this model learns, or other world modeling properties that could be baked into this approach -- otherwise the functionality is quite similar to a nerf
- There could be more analysis of the memory volume produced, what sort of information is store in there, what can we lineary decode, how interpretable is it.

---

> ### Author Rebuttal · Authors · 2025-07-31
>
> Dear Reviewer,
>
> We appreciate the reviewer for the detailed comments and insightful suggestions. We are glad to hear you find that our idea is appealing and our model has significant improvements. Here is the response to your question.
>
> > some evaluations of the physical collisions
>
> Thanks for the suggestions. To evaluate how well our model handles physical collisions, we measure frame-wise similarity using LPIPS when collisions occur. We compare the performance of our model to a baseline that renders videos in the simulation without physical collision handling. As shown in the table, our model is able to accurately simulate future frames after a collision.
>
> |  | LPIPS $\downarrow$ |
> | -------- | -------- |
> | Simulation w/o collision | 0.62 |
> | Ours |  0.25 |
>
>
> > What type of information is stored in this model volume? Can you linearly decode a semantic map of the room?
>
> Our model constructs a volumetric memory representation by populating 3D grids with DINO features. These features capture detailed structural and texture information, which aids in reconstructing the geometry and color of objects in 3D space [1].
>
> Yes. The information stored in our memory can be linearly decoded into a semantic map of the room. This is supported by prior work [2, 3] showing that DINO features are highly effective for dense recognition tasks such as detection and semantic segmentation. Additionally, some works [4] have explored using a similar memory volume as a semantic map for vision-language navigation tasks.
>
>
> [1] Hong, Yicong, et al. "Lrm: Large reconstruction model for single image to 3d." arXiv preprint arXiv:2311.04400 (2023).
>
> [2] Oquab M, Darcet T, Moutakanni T, et al. Dinov2: Learning robust visual features without supervision[J]. arXiv preprint arXiv:2304.07193, 2023.
>
> [3] Ren T, Chen Y, Jiang Q, et al. Dino-x: A unified vision model for open-world object detection and understanding[J]. arXiv preprint arXiv:2411.14347, 2024.
>
> [4] Wang, Zihan, et al. "Gridmm: Grid memory map for vision-and-language navigation." Proceedings of the IEEE/CVF International conference on computer vision. 2023.
>
>
> > What is the sensitivity to voxel size ? How bit of a maximum scene can this approach handle?
>
> Thanks for the suggestion. We evaluated the model with different voxel sizes, as shown in the table. Our results indicate that when the voxel size is 256 or larger, model performance remains stable. However, when the voxel size is set below 256, we observe a slight decrease in performance.
>
> | Voxel Size | LPIPS $\downarrow$ | PSNR $\uparrow$ | SSIM $\uparrow$ |
> | -------- | -------- | -------- | -------- |
> | 128     |    0.20  | 20.6 | 0.73 |
> | 256 (ours) | 0.16 | 22.5 | 0.76 |
> | 384 | 0.16 |  22.2 | 0.77  |
>
> In principle, our approach is capable of handling very large—and even unbounded—scenes. This is achieved by maintaining a large, potentially infinite, grid map and dynamically extracting the relevant local region based on the camera to be fed into the video diffusion model. This design allows our method to scale effectively with scene size.
>
> > How does this compare to existing novel view synthesis and world models out there such as stable video camera?
>
> We compare our model with the stable video camera as shown in the table. We find that stable video camera is good at novel view synthesis, while it also generates new content quite easily without memory.
>
> | Model | LPIPS $\downarrow$ | PSNR $\uparrow$ | SSIM $\uparrow$ |
> | -------- | -------- | -------- | -------- |
> | stable-virtual-camera     |   0.37   | 18.3 |  0.55 |
> | NWM | 0.31  | 17.5 | 0.66 |
> | Ours | 0.16 | 22.5 | 0.76 |
>
> > Does the explicit memory approach only show benefits when the trajectories are very long?
>
> Not only. Our memory approach can improve the 3D spatial consistency of the generated videos. We evaluate the performance of the model for short-horizon generation. The results are shown in the table.
>
> | Short Horizon | LPIPS $\downarrow$ | PSNR $\uparrow$ | SSIM $\uparrow$ |
> | -------- | -------- | -------- | -------- |
> | Ours without memory |  0.20  | 19.8 | 0.74 |
> | Ours | 0.15 | 22.5  | 0.77 |
>
>
> > And in that case how does it compare to simply using an every updating NERF to store a representation of the environment?
>
> Our model not only predicts future observations under action control but also faithfully reconstructs past scenes with a high degree of spatial coherence. NeRF-based approaches are limited in their generative capabilities and cannot synthesize novel elements beyond their original observations. However, constructing and updating NeRF representations is computationally expensive—typically requiring several minutes for a single scene, which makes it infeasible to use NeRF during training or for scalable, interactive applications. In comparison, our method enables more efficient and flexible scene representation and generation, especially in action-conditioned and open-ended settings.

---

> > ### Comment · Reviewer_jqaK · 2025-08-05
> >
> > I would like to thank the author for taking the time to complete this extensive rebuttal. I appreciate the effort that was invested.
> >
> > I think that some of my concerns were addressed but I do not fully understand what the results are saying. Are you saying that Stable Virtual Camera is on par with your method and can do some of the things you claim to be doing? Or are you saying that it is much worse than your method? And if it is the latter do you think this is because your evaluations are ood for SVC. Or are you saying that your method is a universally better NVS method ?
> >
> > It seems to me that with the advances in end to end NVS modeling, that the core advantage of an approach such as yours would be in the ability to make embedding maps and find them useful for som downstream tasks perhaps something with robotics. But it does not seem to be the focus as things stand.
> >
> > I would appreciate a clarification on your stance in regarding to these new SVC results and how you plan on talking about it (as well as some methodological details).
> >
> > I will maintain my score of 4 for now.

---

> ### Author Response · Authors · 2025-08-06
>
> Dear Reviewer,
>
> Thank you for your detailed feedback—we greatly appreciate your insights. We would like to clarify that our core contribution lies not in novel view synthesis, but in addressing a critical limitation of most existing world models: their lack of memory, which hinders an agent’s ability to generate consistent long-horizon plans.
>
> Our proposed memory mechanism is designed to be general, meaning it can be integrated into other world models such as Stable Virtual Camera (SVC). We believe that equipping world models with such persistent memory capabilities will significantly enhance their utility for planning and policy learning.
>
> We would like to clarify that our comparison with Stable Video Camera is just to illustrate and ablate the effect of memory in our rendering setting. While Stable Video Camera is a stronger rendering model, it lacks memory and thus performs worse than our approach on our dataset. We performed a zero-shot evaluation of Stable Virtual Camera, as its training code has not been made open-source. We are willing to explore the integration of our memory framework with SVC in future work.
>
> Best,
>
> The Authors

---

### Note · Authors · 2025-08-12

Dear Reviewers and AC,

We are deeply grateful to the Area Chair and reviewers for your invaluable feedback and the opportunity to provide clarifications. We are encouraged to note that most concerns have been addressed through the discussion period.

We appreciate the reviewers’ recognition of the paper’s key contributions: our method, which explicitly integrates a 3D memory into world models, is shown to be an effective persistent embodied world model, and this model is both beneficial and essential for downstream embodied applications.

We would like to explicitly highlight our corresponding responses to the remaining concerns as follows:

> Real-World Evaluation

We acknowledge that the current evaluation is restricted to synthetic data, and we have explicitly highlighted this limitation in the paper.
We aim to emphasize that the robustness of our model when handling noisy depth inputs suggests potential for real-world evaluation.

> Definitions of the equations

We apologize that we didn’t include the preliminary background to formally define these equations in the paper because of length limitations. In the revised version, we will carefully incorporate this background content to ensure all equations are easily comprehensible.

> Stable Virtual Camera

Our proposed memory mechanism is designed to be general, meaning it can be integrated into other world models such as Stable Virtual Camera (SVC). We believe that equipping world models with such persistent memory capabilities will significantly enhance their utility for planning and policy learning.

Once again, we sincerely thank the reviewers and Area Chair for the time, attention, and consideration dedicated to reviewing our additional remarks.

Best,

The Authors

---

### Decision · Program_Chairs · 2025-09-17

**Decision:**

Accept (poster)

**Comment:**

The paper introduces an embodied world model by incorporating a 3D memory, which leads to better long-horizon generations. After the discussions, the paper received Accept, two Borderline Accept and one Borderline Reject ratings.

Reviewer MRYi maintained their reject rating mainly due to the unjustified claims and lack of clarity. The AC checked the paper, the rebuttal and the discussions and found the paper to be of merit and recommended acceptance as it provides a novel approach for conditioning video generation on a spatial memory. However, the authors must revise the paper according to the comments.

Also, the paper solves a problem similar to slightly older works such as the ones mentioned below. These papers or similar ones should be included in the related work.

[1] Koh et al., Pathdreamer: A World Model for Indoor Navigation\
[2] Kotar et al., ENTL: Embodied Navigation Trajectory Learner